# FEDERATED AVERAGING AS EXPECTACTION-MAXIMIZATION

## ABSTRACT

Federated averaging (`FedAvg`), despite its simplicity, has been the main approach in training neural networks in the federated learning setting. In this work, we show that the algorithmic choices of the `FedAvg` algorithm correspond to optimizing a *single* objective function that involves the global and all of the shard specific models using a *hard* version of the well known Expectation-Maximization (EM) algorithm. As a result, we gain a better understanding of the behavior and design choices of federated averaging while being able to provide interesting connections to recent literature. Based on this view, we further propose `FedSparse`, a version of federated averaging that employs prior distributions to promote model sparsity. In this way, we obtain a procedure that leads to reductions in both server-client and client-server communication costs as well as more efficient models.

## 1 INTRODUCTION

Smart devices have become ubiquitous in today's world and are generating large amounts of potentially sensitive data. Traditionally, such data is transmitted and stored in a central location for training machine learning models. Such methods rightly raise privacy concerns and we seek the means for training powerful models, such as neural networks, without the need to transmit potentially sensitive data. To this end, Federated Learning (FL) (McMahan et al., 2016) has been proposed to train global machine learning models without the need for participating devices to transmit their data to the server. The Federated Averaging (`FedAvg`) (McMahan et al., 2016) algorithm communicates the parameters of the machine learning model instead of the data itself, which is a more private means of communication.

The `FedAvg` algorithm was originally proposed through empirical observations. While it can be shown that it converges (Li et al., 2019), its theoretical understanding in terms of the model assumptions as well the underlying objective function is still not well understood. The first contribution of this work improves our understanding of `FedAvg`; we show that `FedAvg` can be derived by applying the general Expectation-Maximization (EM) framework to a simple hierarchical model. This novel view has several interesting consequences: it sheds light on the algorithmic choices of `FedAvg`, bridges `FedAvg` with meta-learning, connects several extensions of `FedAvg` and provides fruitful ground for future extensions.

Apart from theoretical grounding, the FL scenario poses several practical challenges, especially in the "cross-device" setting (Kairouz et al., 2019) that we consider in this work. In particular, communicating model updates over multiple rounds across a large amount of devices can incur significant communication costs. Communication via the public internet infrastructure and mobile networks is potentially slow and not for free. Equally important, training (and inference) takes place on-device and is therefore restricted by the edge-devices' hardware constraints on memory, speed and heat dissipation capabilities. Therefore, jointly addressing both of these issues is an important step towards building practical FL systems, as also discussed in Kairouz et al. (2019).

Through the novel EM view of `FedAvg` that we introduce, we develop our second contribution, `FedSparse`. `FedSparse` allows for learning sparse models at the client and server via a careful choice of priors within the hierarchical model. As a result, it tackles the aforementioned challenges, since it can simultaneously reduce the overall communication and computation at the client devices. Empirically, `FedSparse` provides better communication-accuracy trade-offs compared to both `FedAvg` as well as methods proposed for similar reasons (Caldas et al., 2018).

## 2 FEDAVG THROUGH THE LENS OF EM

The `FedAvg` algorithm is a simple iterative procedure realized in four simple steps. At the beginning of each round $t$, the server communicates the model parameters, let them be $\mathbf{w}$, to a subset of the devices. The devices then proceed to optimize $\mathbf{w}$, *e.g.*, via stochastic gradient descent, on their respective dataset via a given loss function

$$\mathcal{L}_s(\mathcal{D}_s, \mathbf{w}) := \frac{1}{N_s} \sum_{i=1}^{N_s} L(\mathcal{D}_{si}, \mathbf{w}) \tag{1}$$

where $s$ indexes the device, $\mathcal{D}_s$ corresponds to the dataset at device $s$ and $N_s$ corresponds to its size. After a specific amount of epochs of optimization on $\mathcal{L}_s$ is performed, denoted as $E$, the devices communicate the current state of their parameters, let it be $\phi_s$, to the server. The server then performs an update to its own model by simply averaging the client specific parameters $\mathbf{w}_t = \frac{1}{S} \sum_s \phi_s$.

### 2.1 THE CONNECTION TO EM

We now ask the following question; does the overall algorithm correspond to a specific optimization procedure on a given objective function? Let us consider the following objective function:

$$\arg\max_{\mathbf{w}} \frac{1}{S} \sum_{s=1}^{S} \log p(\mathcal{D}_s|\mathbf{w}), \tag{2}$$

where $\mathcal{D}_s$ corresponds to the shard specific dataset that has $N_s$ datapoints, $p(\mathcal{D}_s|\mathbf{w})$ corresponds to the likelihood of $\mathcal{D}_s$ under the server parameters $\mathbf{w}$. Now consider decomposing each of the shard specific likelihoods as follows:

$$p(\mathcal{D}_s|\mathbf{w}) = \int p(\mathcal{D}_s|\phi_s)p(\phi_s|\mathbf{w})\mathrm{d}\phi_s, \quad p(\phi_s|\mathbf{w}) \propto \exp\left(-\frac{\lambda}{2}\|\phi_s - \mathbf{w}\|^2\right), \tag{3}$$

where we introduced the auxiliary latent variables $\phi_s$, which are the parameters of the local model at shard $s$. The server parameters $\mathbf{w}$ act as "hyperparameters" for the prior over the shard specific parameters and $\lambda$ acts as a regularization strength that prevents $\phi_s$ from moving too far from $\mathbf{w}$.

How can we then optimize the resulting objective in the presence of these latent variables $\phi_s$? The traditional way to optimize such objectives is through Expectation-Maximization (EM). EM consists of two steps, the E-step where we form the posterior distribution over these latent variables

$$p(\phi_s|\mathcal{D}_s, \mathbf{w}) = \frac{p(\mathcal{D}_s|\phi_s)p(\phi_s|\mathbf{w})}{p(\mathcal{D}_s|\mathbf{w})}, \tag{4}$$

and the M-step where we maximize the probability of $\mathcal{D}_s$ w.r.t. the parameters of the model $\mathbf{w}$ by marginalizing over this posterior

$$\arg\max_{\mathbf{w}} \frac{1}{S} \sum_s \mathbb{E}_{p(\phi_s|\mathcal{D}_s, \mathbf{w}_{\mathrm{old}})}[\log p(\mathcal{D}_s|\phi_s) + \log p(\phi_s|\mathbf{w})]$$

$$= \arg\max_{\mathbf{w}} \frac{1}{S} \sum_s \mathbb{E}_{p(\phi_s|\mathcal{D}_s, \mathbf{w}_{\mathrm{old}})}[\log p(\phi_s|\mathbf{w})]. \tag{5}$$

If we perform a single gradient step for $\mathbf{w}$ in the M-step, this procedure corresponds to doing gradient ascent on the original objective, a fact we show in Appendix D.

When posterior inference is intractable, hard-EM is sometimes employed. In this case we make "hard" assignment for the latent variables $\phi_s$ in the E-step by approximating $p(\phi_s|\mathcal{D}_s, \mathbf{w})$ with its most probable point, *i.e.*

$$\phi_s^* = \arg\max_{\phi_s} \frac{p(\mathcal{D}_s|\phi_s)p(\phi_s|\mathbf{w})}{p(\mathcal{D}_s|\mathbf{w})} = \arg\max_{\phi_s} \log p(\mathcal{D}_s|\phi_s) + \log p(\phi_s|\mathbf{w}). \tag{6}$$

This is usually easier to do as we can use techniques such as stochastic gradient ascent. Given these hard assignments, the M-step then corresponds to another simple maximization

$\arg\max_{\mathbf{w}} \frac{1}{S} \sum_s \log p(\phi_s^* | \mathbf{w})$. As a result, hard-EM corresponds to a block coordinate ascent type of algorithm on the following objective function

$$\arg\max_{\phi_{1:S},\mathbf{w}} \frac{1}{S} \sum_s (\log p(\mathcal{D}_s|\phi_s) + \log p(\phi_s|\mathbf{w})), \qquad (7)$$

where we alternate between optimizing $\phi_{1:S}$ and $\mathbf{w}$ while keeping the other fixed.

How is this framework the same as FedAvg? By letting $\lambda \to 0$ in Eq. 3 it is clear that the hard assignments in the E-step mimic the process of optimizing a local model on the data of each shard. In fact, even by optimizing the model locally with stochastic gradient ascent for a fixed number of iterations with a given learning rate we implicitly assume a specific prior over the parameters; for linear regression, this prior is a Gaussian centered at the initial value of the parameters (Santos, 1996) whereas for non-linear models it bounds the distance from the initial point. After obtaining $\phi_s^*$ the M-step then corresponds to $\arg\max_{\mathbf{w}} \mathcal{L}_r := \frac{1}{S} \sum_s -\frac{\lambda}{2}\|\phi_s^* - \mathbf{w}\|^2$, and we can easily find a closed form solution by setting the derivative of the objective w.r.t. $\mathbf{w}$ to zero and solving for $\mathbf{w}$:

$$\frac{\partial \mathcal{L}_r}{\partial \mathbf{w}} = 0 \Rightarrow \frac{\lambda}{S}\sum_s (\phi_s^* - \mathbf{w}) = 0 \Rightarrow \mathbf{w} = \frac{1}{S}\sum_s \phi_s^*. \qquad (8)$$

It is easy to see that the optimal solution for $\mathbf{w}$ given $\phi_{1:S}^*$ is the same as the one from FedAvg. Of course, FedAvg does not optimize the local parameters $\phi_s$ to convergence at each round, so one might wonder whether the correspondence to EM is still valid. It turns out that the alternating procedure of EM corresponds to block coordinate ascent on a single objective function, the variational lower bound of the marginal log-likelihood (Neal & Hinton, 1998) of a given model. More specifically for our setting, we can see that the EM iterations perform block coordinate ascent to optimize:

$$\arg\max_{\mathbf{w}_{1:S},\mathbf{w}} \frac{1}{S} \sum_s \mathbb{E}_{q_{\mathbf{w}_s}(\phi_s)}\big[\log p(\mathcal{D}_s|\phi_s) + \log p(\phi_s|\mathbf{w})\big] + H[q_{\mathbf{w}_s}(\phi_s)] \qquad (9)$$

where $\mathbf{w}_s$ are the parameters of the variational approximation to the posterior distribution $p(\phi_s|\mathcal{D}_s, \mathbf{w})$ and $H[q]$ corresponds to the entropy of the $q$ distribution. To obtain the procedure of FedAvg we can use a (numerically) deterministic distribution for $\phi_s$, $q_{\mathbf{w}_s}(\phi_s) := \mathcal{N}(\mathbf{w}_s, \epsilon\mathbf{I})$. This leads us to the same objective as in Eq. 7, since the expectation concentrates on a single term and the entropy of $q_{\mathbf{w}_s}(\phi_s)$ becomes a constant independent of the optimization. In this case, the optimized value for $\phi_s$ after a fixed number of steps corresponds to the $\mathbf{w}_s$ of the variational approximation.

It is interesting to contrast recent literature under the lens of this framework. Optimizing the same hierarchical model with hard-EM but with a non-trivial $\lambda$ results into the same procedure that was proposed by Li et al. (2018). Furthermore, using the difference of the local parameters to the global parameters as a "gradient" (Reddi et al., 2020) is equivalent to hard-EM on the same model where in the M-step we take a single gradient step. In addition, this view makes precise the idea that FedAvg is a meta-learning algorithm (Jiang et al., 2019); the underlying hierarchical model we optimize is similar to the ones used in meta-learning (Grant et al., 2018; Chen et al., 2019).

How can we then use this novel view of FedAvg to our advantage? The most straightforward way is to use an alternative prior which would result into different behaviours in local training and server side updating. For example, one could use a Laplace prior, which would result into the server selecting the median instead of averaging, or a mixture of Gaussians prior, which would result into training an ensemble of models at the server. In order to tackle the communication and computational costs, which is important for "cross-device" FL, we chose a sparsity inducing prior, namely the spike and slab prior. We describe the resulting algorithm, FedSparse, in the next section.

## 3 THE FEDSPARSE ALGORITHM: SPARSITY IN FEDERATED LEARNING

Encouraging sparsity in FL has two main advantages; the model becomes smaller and less resource intensive to train and furthermore, it cuts down on communication costs as the pruned parameters do not need to be communicated. The golden standard for sparsity in probabilistic models is the spike and slab (Mitchell & Beauchamp, 1988) prior. It is a mixture of two components, a delta spike at zero, $\delta(0)$, and a continuous distribution over the real line, *i.e.* the slab. More specifically, by adopting a Gaussian slab for each local parameter $\phi_{si}$ we have that

$$p(\phi_{si}|\boldsymbol{\theta}_i, \mathbf{w}_i) = (1 - \boldsymbol{\theta}_i)\delta(0) + \boldsymbol{\theta}_i\mathcal{N}(\phi_{si}|\mathbf{w}_i, 1/\lambda), \qquad (10)$$

or equivalently as a hierarchical model

$$p(\boldsymbol{\phi}_{si}|\boldsymbol{\theta}_i, \mathbf{w}_i) = \sum_{\mathbf{z}_{si}} p(\mathbf{z}_{si}|\boldsymbol{\theta}_i)p(\boldsymbol{\phi}_{si}|\mathbf{z}_{si}, \mathbf{w}_i), \quad p(\mathbf{z}_{si}) = \text{Bern}(\boldsymbol{\theta}_i), \tag{11}$$

$$p(\boldsymbol{\phi}_{si}|\mathbf{z}_{si} = 1, \mathbf{w}_i) = \mathcal{N}(\boldsymbol{\phi}_{si}|\mathbf{w}_i, 1/\lambda), \quad p(\boldsymbol{\phi}_{si}|\mathbf{z}_{si} = 0) = \delta(0), \tag{12}$$

where $\mathbf{z}_{si}$ plays the role of a "gating" variable that switches on or off the parameter $\boldsymbol{\phi}_{si}$. Now consider using this distribution for the prior over the parameters in the federated setting. $\mathbf{w}, \boldsymbol{\theta}$ will be the server side model weights and probabilities of the binary gates. In order to stay close to the `FedAvg` paradigm of simple point estimation and since approximate inference for complex posteriors, such as those that arise in neural networks, is still an open problem, we will perform hard-EM in order to optimize $\mathbf{w}, \boldsymbol{\theta}$. By using approximate distributions $q_{\mathbf{w}_s}(\boldsymbol{\phi}_s|\mathbf{z}_s), q_{\boldsymbol{\pi}_s}(\mathbf{z}_s)$, the variational lower bound for this model becomes

$$\arg\max_{\mathbf{w}_{1:S}, \mathbf{w}, \boldsymbol{\pi}_{1:S}, \boldsymbol{\theta}} \frac{1}{S} \sum_s \mathbb{E}_{q_{\boldsymbol{\pi}_s}(\mathbf{z}_s)q_{\mathbf{w}_s}(\boldsymbol{\phi}_s|\mathbf{z}_s)} \big[ \log p(\mathcal{D}_s|\boldsymbol{\phi}_s) + \log p(\boldsymbol{\phi}_s|\mathbf{w}, \mathbf{z}_s) + \log p(\mathbf{z}_s|\boldsymbol{\theta})$$
$$- \log q_{\mathbf{w}_s}(\boldsymbol{\phi}_s|\mathbf{z}_s) \big] + H[q_{\boldsymbol{\pi}_s}(\mathbf{z}_s)]. \tag{13}$$

For the shard specific weight distributions, as they are continuous, we will use $q_{\mathbf{w}_s}(\boldsymbol{\phi}_{si}|\mathbf{z}_{si} = 1) := \mathcal{N}(\mathbf{w}_{si}, \epsilon), q(\boldsymbol{\phi}_{si}|\mathbf{z}_{si} = 1) := \mathcal{N}(0, \epsilon)$ with $\epsilon \approx 0$ which will be, numerically speaking, deterministic. For the gating variables, as they are binary, we will use $q_{\boldsymbol{\pi}_{si}}(\mathbf{z}_{si}) := \text{Bern}(\boldsymbol{\pi}_{si})$ with $\boldsymbol{\pi}_{si}$ being the probability of activating local gate $\mathbf{z}_{si}$. In order to do hard-EM for the binary variables, we will remove the entropy term for the $q_{\boldsymbol{\pi}_s}(\mathbf{z}_s)$ from the aforementioned bound as this will encourage the approximate distribution to move towards the most probable value for $\mathbf{z}_s$. Furthermore, by relaxing the spike at zero to a Gaussian with precision $\lambda_2$, i.e., $p(\boldsymbol{\phi}_{si}|\mathbf{z}_{si} = 0) = \mathcal{N}(0, 1/\lambda_2)$, and by plugging in the appropriate expressions into Eq. 13 we can show that the local and global objectives will be

$$\arg\max_{\mathbf{w}_s, \boldsymbol{\pi}_s} \mathcal{L}_s(\mathcal{D}_s, \mathbf{w}, \boldsymbol{\theta}, \mathbf{w}_s, \boldsymbol{\pi}_s) := \mathbb{E}_{q_{\boldsymbol{\pi}_s}(\mathbf{z}_s)} \left[ \sum_i^{N_s} L(\mathcal{D}_{si}, \mathbf{w}_s \odot \mathbf{z}_s) \right] - \frac{\lambda}{2} \sum_j \boldsymbol{\pi}_{sj}(\mathbf{w}_{sj} - \mathbf{w}_j)^2$$
$$- \lambda_0 \sum_j \boldsymbol{\pi}_{sj} + \sum_j (\boldsymbol{\pi}_{sj} \log \boldsymbol{\theta}_j + (1 - \boldsymbol{\pi}_{sj}) \log(1 - \boldsymbol{\theta}_j)) + C, \tag{14}$$

$$\arg\max_{\mathbf{w}, \boldsymbol{\theta}} \mathcal{L} := \frac{1}{S} \sum_{s=1}^S \mathcal{L}_s(\mathcal{D}_s, \mathbf{w}, \boldsymbol{\theta}, \mathbf{w}_s, \boldsymbol{\pi}_s) \tag{15}$$

respectively, where $\lambda_0 = \frac{1}{2} \log \frac{\lambda_2}{\lambda}$ and $C$ is a constant independent of the variables to be optimized. The derivation can be found at Appendix E. It is interesting to see that the final objective at each shard intuitively tries to find a trade-off between four things: 1) explaining the local dataset $\mathcal{D}_s$, 2) having the local weights close to the server weights (regulated by $\lambda$), 3) having the local gate probabilities close to the server probabilities and 4) reducing the local gate activation probabilities so as to prune away a parameter (regulated by $\lambda_0$). The latter is an $L_0$ regularization term, akin to the one proposed by Louizos et al. (2017).

Now let us consider what happens at the server after the local shard, through some procedure, optimized $\mathbf{w}_s$ and $\boldsymbol{\pi}_s$. Since the server loss for $\mathbf{w}, \boldsymbol{\theta}$ is the sum of all local losses, the gradient for each of the parameters will be

$$\frac{\partial \mathcal{L}}{\partial \mathbf{w}} = \sum_s \lambda \boldsymbol{\pi}_s(\mathbf{w}_s - \mathbf{w}), \qquad \frac{\partial \mathcal{L}}{\partial \boldsymbol{\theta}} = \sum_s \left( \frac{\boldsymbol{\pi}_s}{\boldsymbol{\theta}} - \frac{1 - \boldsymbol{\pi}_s}{1 - \boldsymbol{\theta}} \right). \tag{16}$$

Setting these derivatives to zero, we see that the stationary points are

$$\mathbf{w} = \frac{1}{\sum_j \boldsymbol{\pi}_j} \sum_s \boldsymbol{\pi}_s \mathbf{w}_s, \qquad \boldsymbol{\theta} = \frac{1}{S} \sum_s \boldsymbol{\pi}_s \tag{17}$$

i.e., a weighted average of the local weights and an average of the local probabilities of keeping these weights. Therefore, since the $\boldsymbol{\pi}_s$ are being optimized to be sparse through the $L_0$ penalty, the server probabilities $\boldsymbol{\theta}$ will also become small for the weights that are used by only a small fraction of the shards. As a result, to obtain the final sparse architecture, we can prune the weights whose corresponding server inclusion probabilities $\boldsymbol{\theta}$ are less than a threshold, e.g., 0.1. It should be noted that the sums and averages of Eq. 16, 17 respectively can be easily approximated by subsampling a small set of clients $S'$ from $S$. Therefore we do not have to consider all of the clients at each round, which would be prohibitive for the "cross-device" setting of FL.

### 3.1 REDUCING THE COMMUNICATION COST

The framework described so far allow us to learn a more efficient model. We now discuss how we can use it in order to cut down both download and upload communication costs during training.

**Reducing client to server communication cost** In order to reduce the client to server cost we will communicate sparse samples from the local distributions instead of the distributions themselves; in this way we do not have to communicate the zero values of the parameter vector. This leads to large savings, while still keeping the server gradient unbiased. More specifically, we can express the gradients and stationary points for the server weights and probabilities as follows

$$\frac{\partial \mathcal{L}}{\partial \mathbf{w}} = \sum_s \lambda \mathbb{E}_{q_{\boldsymbol{\pi}_s}(\mathbf{z}_s)} \left[ \mathbf{z}_s (\mathbf{w}_s - \mathbf{w}) \right], \quad \mathbf{w} = \mathbb{E}_{q_{\boldsymbol{\pi}_{1:S}}(\mathbf{z}_{1:S})} \left[ \frac{1}{\sum_j \mathbf{z}_j} \sum_s \mathbf{z}_s \mathbf{w}_s \right], \tag{18}$$

$$\frac{\partial \mathcal{L}}{\partial \boldsymbol{\theta}} = \sum_s \mathbb{E}_{q_{\boldsymbol{\pi}_s}(\mathbf{z}_s)} \left[ \frac{\mathbf{z}_s}{\boldsymbol{\theta}} - \frac{1 - \mathbf{z}_s}{1 - \boldsymbol{\theta}} \right], \quad \boldsymbol{\theta} = \frac{1}{S} \sum_s \mathbb{E}_{q_{\boldsymbol{\pi}_s}(\mathbf{z}_s)} \left[ \mathbf{z}_s \right]. \tag{19}$$

As a result, we can then communicate from the client only the subset of the local weights $\hat{\mathbf{w}}_s$ that are non-zero in $\mathbf{z}_s \sim q_{\boldsymbol{\pi}_s}(\mathbf{z}_s)$, $\hat{\mathbf{w}}_s = \mathbf{w}_s \odot \mathbf{z}_s$, and the server can infer the state of $\mathbf{z}_s$ by inspecting which parameters were omitted. Having access to those samples, the server can then form 1-sample stochastic estimates of either the gradients or the stationary points for $\mathbf{w}, \boldsymbol{\theta}$. Notice that this is a way to reduce communication without adding bias in the gradients of the original objective. In case that we are willing to incur extra bias, we can further use techniques such as quantization (Amiri et al., 2020) and top-k gradient selection (Lin et al., 2017) to reduce communication even further. Such approaches are left for future work.

**Reducing the server to client communication cost** The server needs to communicate to the clients the updated distributions at each round. Unfortunately, for simple unstructured pruning, this doubles the communication cost as for each weight $\mathbf{w}_i$ there is an associated $\boldsymbol{\theta}_i$ that needs to be sent to the client. To mitigate this effect we will employ structured pruning, which introduces a single additional parameter for each group of weights. For groups of moderate sizes, *e.g.*, the set of weights of a given convolutional filter, the extra overhead is small. We can also take the communication cost reductions one step further if we allow for some bias in the optimization procedure; we can prune the global model during training after every round and thus send to each of the clients only the subset of the model that has survived. Notice that this is easy to do and does not require any data at the server. The inclusion probabilities $\boldsymbol{\theta}$ are available at the server, so we can remove the parameters that have $\boldsymbol{\theta}$ less than a threshold, *e.g.* $0.1$. This can lead to large reductions in communication costs, especially once the model becomes sufficiently sparse.

### 3.2 FEDSPARSE IN PRACTICE

**Local optimization** While optimizing for $\mathbf{w}_s$ locally is straightforward to do with gradient based optimizers, $\boldsymbol{\pi}_s$ is more tricky, as the expectation over the binary variables $\mathbf{z}_s$ in Eq. 14 is intractable to compute in closed form and using Monte-Carlo integration does not yield reparametrizable samples. To circumvent these issues, we rewrite the objective in an equivalent form and use the hard-concrete relaxation from (Louizos et al., 2017), which can allow for the straightforward application of gradient ascent. We provide the details in Appendix F. When the client has to communicate to the server, we propose to form $\hat{\mathbf{w}}_s$ by sampling from the zero-temperature relaxation, which yields exact binary samples. Furthermore, at the beginning of each round, following the practice of `FedAvg`, the participating clients initialize their approximate posteriors to be equal to the priors that were communicated from the server. Empirically, we found that this resulted in better global accuracy.

**Parametrization of the probabilities** Since there have been evidence that such optimization based pruning can be inferior to simple magnitude based pruning (Gale et al., 2019), we take an approach that combines the two and reminisces the recent work of Azarian et al. (2020). We parametrize the probabilities $\boldsymbol{\theta}, \boldsymbol{\pi}_s$ as a function of the model weights and magnitude based thresholds that regulate how active a parameter can be. More specifically, we use the following parametrization

$$\theta_g := \sigma \left( \frac{\|\mathbf{w}_g\|_2 - \tau_g}{T} \right), \quad \pi_{sg} := \sigma \left( \frac{\|\mathbf{w}_{sg}\|_2 - \tau_{sg}}{T} \right), \tag{20}$$

where the subscript $g$ denotes the group, $\sigma(\cdot)$ is the sigmoid function, $\tau_g, \tau_{sg}$ are the global and client specific thresholds for a given group $g$ and $T$ is a temperature hyperparameter. Following Azarian et al. (2020) we also "detach" the gradient of the weights through $\boldsymbol{\theta}, \boldsymbol{\pi}_s$, to avoid decreasing the probabilities by just shrinking the weights. With this parametrization we lose the ability to get a closed form solution for the server thresholds, but nonetheless we can still perform gradient based optimization at the server by using the chain rule. For a positive threshold, we use a parametrization in terms of a softplus function, $i.e.$, $\boldsymbol{\tau} = \log(1 + \exp(\mathbf{v}))$ where $\mathbf{v}$ is the learnable parameter.

---

**Algorithm 1** The server side algorithm for `FedSparse` (assuming weight sparsity for simplicity). $\sigma(\cdot)$ is the sigmoid function, $\epsilon$ is the threshold for pruning.

---

Initialize $\mathbf{v}$ and $\mathbf{w}$
**for** round $t$ in $1, \ldots T$ **do**
    $\boldsymbol{\tau} \leftarrow \log(1 + \exp(\mathbf{v}))$
    $\boldsymbol{\theta} \leftarrow \sigma\left((|\mathbf{w}| - \boldsymbol{\tau})/T\right)$
    $\mathbf{w} \leftarrow \mathbb{I}[\boldsymbol{\theta} > \epsilon]\mathbf{w}$           $\triangleright$ prune global model
    Initialize $\nabla_{\mathbf{w}}^t = \mathbf{0}, \nabla_{\mathbf{v}}^t = \mathbf{0}$
    **for** $s$ in random subset of the clients **do**
        $\hat{\mathbf{w}}_s^t \leftarrow \text{CLIENT}(s, \mathbf{w}, \mathbf{v})$
        $\mathbf{z}_s \leftarrow \mathbb{I}[\hat{\mathbf{w}}_s^t \neq 0]$
        $\nabla_{\mathbf{w}}^t \mathrel{+}= \mathbf{z}_s(\hat{\mathbf{w}}_s^t - \mathbf{w})$
        $\nabla_{\mathbf{v}}^t \mathrel{+}= -\left(\mathbf{z}_s(1 - \boldsymbol{\theta}) - (1 - \mathbf{z}_s)\boldsymbol{\theta}\right)\sigma(\mathbf{v})/T$
    **end for**
    $\mathbf{w}^{t+1}, \mathbf{v}^{t+1} \leftarrow \text{ADAM}(\nabla_{\mathbf{w}}^t), \text{ADAMAX}(\nabla_{\mathbf{v}}^t)$
**end for**

---

**Algorithm 2** The client side algorithm for `FedSparse`.

---

Get $\mathbf{w}, \mathbf{v}$ from the server
$\boldsymbol{\theta} \leftarrow \sigma\left((|\mathbf{w}| - \boldsymbol{\tau})/T\right)$
$\mathbf{w}_s, \mathbf{v}_s \leftarrow \mathbf{w}, \mathbf{v}$
**for** epoch $e$ in $1, \ldots, E$ **do**
    **for** batch $b \in B$ **do**
        $\boldsymbol{\tau}_s \leftarrow \log(1 + \exp(\mathbf{v}_s))$
        $\boldsymbol{\pi}_s \leftarrow \sigma\left((|\mathbf{w}_s| - \boldsymbol{\tau}_s)/T\right)$
        $L_s \leftarrow \mathcal{L}_s(b, \mathbf{w}, \boldsymbol{\theta}, \mathbf{w}_s, \boldsymbol{\pi}_s)$
        $\mathbf{w}_s \leftarrow \text{SGD}(\nabla_{\mathbf{w}_s} L_s)$
        $\mathbf{v}_s \leftarrow \text{ADAMAX}(\nabla_{\mathbf{v}_s} L_s)$
    **end for**
**end for**
$\boldsymbol{\pi}_s \leftarrow \sigma\left((|\mathbf{w}_s| - \boldsymbol{\tau}_s)/T\right)$
$\mathbf{z}_s \sim q_{\boldsymbol{\pi}_s}(\mathbf{z}_s)$
**return** $\mathbf{z}_s \odot \mathbf{w}_s$

---

## 4 RELATED WORK

Both contributions of this work share similarities with several recent works in federated learning. `FedProx` (Li et al., 2018) proposed to add a proximal term to the local objective at each shard, so that it prevents the local models from drifting too far from the global model. Through the EM view of `FedAvg` we show how such a local objective arises if we use a non-trivial precision for the Gaussian prior over the local parameters. Furthermore, `FedAvg` has been advocated to be a meta-learning algorithm in Jiang et al. (2019); with the EM view we make this claim precise and show that the underlying hierarchical model that `FedAvg` optimizes is the same as the models used in several meta-learning works (Grant et al., 2018; Chen et al., 2019). Furthermore, by performing a single gradient step for the M-step in the EM view of `FedAvg` we see that we arrive at a procedure that has been previously explored both in a meta-learning context with the Reptile algorithm (Nichol et al., 2018), as well as the federated learning context with the "generalized" `FedAvg` (Reddi et al., 2020). One important difference between meta-learning and `FedAvg` is that the latter maximizes the average, across shards, marginal-likelihood in order to find the server / global parameters whereas meta-learning methods usually optimize the global parameters such that the finetuned model perform well on the local validation set. Exploring such parameter estimation methods, as, $e.g.$, in Chen et al. (2019), in the federated scenario and how these relate to existing approaches that merge meta-learning with federated learning, $e.g.$, Fallah et al. (2020), is an interesting avenue for future work. Finally, the EM view we provide here for `FedAvg` can also provide a novel perspective for optimization works that improve model performance via model replicas (Zhang et al., 2019; Pittorino et al., 2020).

Reducing the communication costs is a well known and explored topic in federated learning. `FedSparse` has close connections to federated dropout (Caldas et al., 2018), as the latter can be understood via a similar hierarchical model, where gates $\mathbf{z}$ are global and have a fixed probability $\boldsymbol{\theta}$ for both the prior and the approximate posterior. Then at each round and for each client, the server samples these gates (due to the expectation), collects the submodels and sends those to be optimized at the shard. Compared to federated dropout, `FedSparse` allow us to optimize the dropout rates to the data, such that they satisfy a given accuracy / sparsity trade-off, dictated by the hyperparameter $\lambda_0$. Another benefit we gain from the EM view of `FedAvg` is that it makes it clear that the server

can perform gradient based optimization. As a result, we can harvest the large literature on efficient distributed optimization (Lin et al., 2017; Bernstein et al., 2018; Wangni et al., 2018; Yu et al., 2019), which involves gradient quantization, sparsification and more general compression. On this front, there have been also other works that aim to reduce the communication cost in FL via such approaches (Sattler et al., 2019; Han et al., 2020). In general, such approaches can be orthogonal to `FedSparse` and exploring how they can be incorporated is a promising avenue for future research.

## 5  EXPERIMENTS

We verify in three tasks whether the `FedSparse` procedure leads to similar or better global models compared to `FedAvg` while providing reductions in communication costs and efficient models. As a baseline that also reduces communication costs by sparsifying the model, we consider the federated dropout procedure from Caldas et al. (2018), which we refer to as `FedDrop`. For each of the three tasks, we present the results for `FedSparse` with regularization strengths that target three sparsity levels: low, mid and high. For the `FedDrop` baseline, we experiment with multiple combinations of dropout probabilities for the convolutional and fully connected layers. For each of these tasks we report the setting that performs best in term of accuracy / communication trade-off.

The first task we consider is a federated version of CIFAR10 classification; we partition the data among 100 shards in a non-i.i.d. way by following Hsu et al. (2019). For the model we employ a LeNet-5 convolutional architecture (LeCun et al., 1998) with the addition of dropout(0.1) for the second convolutional layer and dropout(0.3) for the first fully connected layer in order to prevent overfitting locally at the shard. We optimize the model for 1k communication rounds. For the second task we consider the 500 shard federated version of CIFAR100 classification from Reddi et al. (2020). For the model we use a ResNet20, where we replace the batch normalization layers with group normalization, following Reddi et al. (2020), and we optimize for 6k rounds. For the final task we considered the non-i.i.d. Femnist classification and we use the same configuration as CIFAR10 but we optimize the model for 6k rounds. More details can be found at Appendix A.

We evaluate `FedSparse` and the baselines on two metrics that highlight the tradeoffs between accuracy and communication costs. On both metrics the x-axis represents the total communication cost incurred up until that point and the y-axis represents two distinct model accuracies. The first one corresponds to the accuracy of the global model on the union of the shard test sets, whereas the second one corresponds to the average accuracy of the shard specific "local models" on the shard specific test sets. The "local model" on each shard is the model configuration that the shard last communicated to the server, and serves as a proxy for the personalized model performance on each shard. The later metric is motivated from the meta-learning (Jiang et al., 2019) and EM view of federated averaging, and corresponds to using the local posteriors for prediction on the local test set instead of the server side priors.

### 5.1  EXPERIMENTAL RESULTS

The results from our experiments can be found in the following table and figures. Overall, we observed that the `FedSparse` models achieve their final sparsity ratios early in training, i.e., after 30-50 rounds, which quickly reduces the communication costs for each round (Appendix B).

Table 1: Average global, local test-set accuracies across clients in %, along with total communications costs in GB and sparsity of the final model for Cifar10, Cifar100 and Femnist. We report the average over the last 10 evaluations.

| Method | Cifar 10 | | | | Cifar 100 | | | | Femnist | | | |
|---|---|---|---|---|---|---|---|---|---|---|---|---|
| | G.Acc. | L.Acc. | Comm. | Spars. | G.Acc. | L.Acc. | Comm. | Spars. | G.Acc. | L.Acc. | Comm. | Spars. |
| `FedAvg` | 69.97 | 86.71 | 65 | - | 41.11 | 61.61 | 123 | - | 85.62 | 90.81 | 272 | - |
| `FedDrop` | 71.54 | 86.86 | 35 | - | 37.65 | 58.02 | 112 | - | 85.23 | 89.37 | 174 | - |
| `FedSparse, low` | 70.30 | 87.65 | 52 | 20.1 | 40.69 | 61.31 | 113 | 2.2 | 85.03 | 90.82 | 270 | 1.0 |
| `FedSparse, mid` | 70.30 | 87.54 | 36 | 45.7 | 31.94 | 54.83 | 59 | 51.0 | 84.83 | 90.61 | 145 | 47.0 |
| `FedSparse, high` | 68.46 | 87.17 | 26 | 62.4 | 24.68 | 49.58 | 45 | 62.9 | 84.13 | 89.68 | 95 | 65.5 |

We can see that for CIFAR 10, the `FedSparse` models with medium (~45%) and high (~62%) sparsity outperform all other methods for small communications budgets on the global accuracy front, but are eventually surpassed by `FedDrop` on higher budgets. However, on the local accuracy front, we see that the `FedSparse` models Pareto dominate both baselines, achiev-

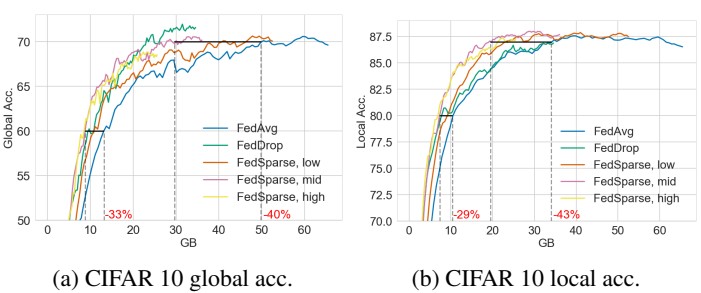

(a) CIFAR 10 global acc.

(b) CIFAR 10 local acc.

ing, *e.g.*, 87% local accuracy with 43% less communication compared to `FedAvg`. Overall, judging the final performance only, we see that `FedDrop` reaches the best accuracy on the global model, but `FedSparse` reaches the best accuracy in the local models.

On CIFAR 100, the differences are less pronounced, as the models did not fully converge for the maximum number of rounds we use. Nevertheless, we still observe similar patterns; for small communication budgets, the sparser models are better for both the global and local accuracy as, *e.g.*, they can reach 32% global accuracy while requiring 13% less communication than

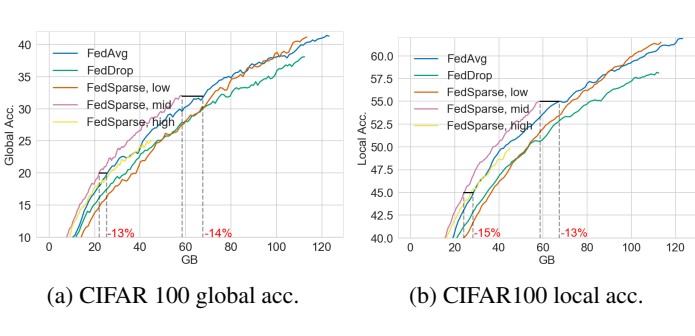

(a) CIFAR 100 global acc.

(b) CIFAR100 local acc.

`FedAvg`.

Finally for Femnist we observe the most differences, as the Femnist task is also more communication intensive due to having 3.5k shards. We see that the `FedSparse` algorithm Pareto dominates both `FedDrop` and `FedAvg` and, more specifically, in the high sparsification setting, it can reach 84% global accuracy and 89% local accuracy while requiring 41% and 51% less com-

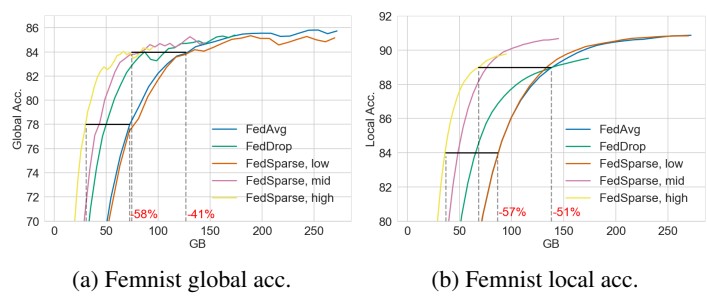

(a) Femnist global acc.

(b) Femnist local acc.

munication compared to `FedAvg` respectively. Judging by the final accuracy; both `FedAvg` and `FedSparse` with the low setting reached similar global and local model performance, which is to be expected given that that particular `FedSparse` setting lead to only 1% sparsity.

## 6  CONCLUSION

In this work, we showed how the `FedAvg` algorithm, the standard in federated learning, corresponds to applying a variant of the well known EM algorithm to a simple hierarchical model. Through this perspective, we bridge several recent works on federated learning as well as connect `FedAvg` to meta-learning. As a straightforward extension stemming from this view, we proposed `FedSparse`, a generalization of `FedAvg` with sparsity inducing priors. Empirically, we showed that `FedSparse` can learn sparse neural networks which, besides being more efficient, can also significantly reduce the communication costs without decreasing performance.

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

## APPENDIX

## A EXPERIMENTAL DETAILS

For all of the three tasks we randomly select 10 clients without replacement in a given round but with replacement across rounds. For the local optimizer of the weights we use stochastic gradient descent with a learning rate of 0.05, whereas for the global optimizer we use Adam (Kingma & Ba, 2014) with the default hyperparameters provided in (Kingma & Ba, 2014). For the pruning thresholds in `FedSparse` we used the Adamax (Kingma & Ba, 2014) optimizer with $1e-3$ learning rate at the shard level and the Adamax optimizer with $1e-2$ learning rate at the server. For all three of the tasks we used $E = 1$ with a batch size of 64 for CIFAR10 and 20 for CIFAR100 and Femnist. It should be noted that for all the methods we performed gradient based optimization using the difference gradient for the weights (Reddi et al., 2020) instead of averaging.

For the `FedDrop` baseline, we used a very small dropout rate of 0.01 for the input and output layer and tuned the dropout rates for convolutional and fully connected layers separately in order to optimize the accuracy / communication tradeoff. For convolutional layers we considered rates in $\{0.1, 0.2, 0.3\}$ whereas for the fully connected layers we considered rates in $\{0.1, 0.2, 0.3, 0.4, 0.5\}$. For CIFAR10 we did not employ the additional dropout noise at the shard level, since we found that it was detrimental for the `FedDrop` performance. Furthermore, for Resnet20 on CIFAR100 we did not apply federated dropout at the output layer. For CIFAR10 the best performing dropout rates were 0.1 for the convolutional and 0.5 for the fully connected, whereas for CIFAR100 it was 0.1 for the convolutional. For Femnist, we saw that a rate of 0.2 for the convolutional and a rate of 0.4 for the fully connected performed better.

For `FedSparse`, we initialized $\mathbf{v}$ such that the thresholds $\boldsymbol{\tau}$ lead to $\boldsymbol{\theta} = 0.99$ initially, *i.e.* we started from a dense model. The temperature for the sigmoid in the parameterization of the probabilities was set to $T = 0.001$. Furthermore, we downscaled the cross-entropy term between the client side

probabilities, $\boldsymbol{\pi}_s$, and the server side probabilities, $\boldsymbol{\theta}$ by mutltiplying it with $1e-4$. Since at the beginning of each round we were always initializing $\boldsymbol{\pi}_S = \boldsymbol{\theta}$ and we were only optimizing for a small number of steps before synchronizing, we found that the full strength of the cross-entropy was not necessary. Furthermore, for similar reasons, *i.e.* we set $\mathbf{w}_s = \mathbf{w}$ at the beginning of each round, we also used $\lambda = 0$ for the drift term $\frac{\lambda}{2}\boldsymbol{\pi}_{sj}(\mathbf{w}_s - \mathbf{w})^2$. The remaining hyperparameter $\lambda_0$ dictates how sparse the final model will be. For the LeNet-5 model the $\lambda_0$'s we report are $\{5e-7, 5e-6, 5e-5\}$ for the "low", "mid" and "high" settings respectively, which were optimized for CIFAR10 and used as-is for Femnist. For CIFAR100 and Resnet20, we did not perform any pruning for the output layer and the $\lambda_0$'s for the "low", "mid" and "high" settings were $\{5e-7, 5e-6, 5e-5\}$ respectively. These were chosen so that we obtain models with comparable sparsity ratios as the one on CIFAR10.

## B    EVOLUTION OF SPARSITY

We show the evolution of the sparsity ratios for all tasks and configurations in the following plot. We can see that in all settings the model attains its final sparsity quite early in training.

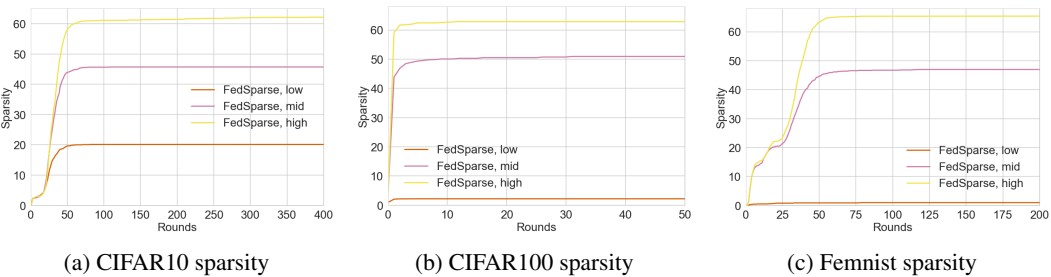

(a) CIFAR10 sparsity                    (b) CIFAR100 sparsity                    (c) Femnist sparsity

## C    ADDITIONAL RESULTS

**Convergence plots in terms of communication rounds.**    In order to understand whether the extra noise is detrimental to the convergence speed of `FedSparse`, we plot the validation accuracy in terms of communication rounds for all tasks and baselines. As it can be seen, there is no inherent difference before `FedSparse` starts pruning. This happens quite early in training for CIFAR 100 thus it is there where we observe the most differences.

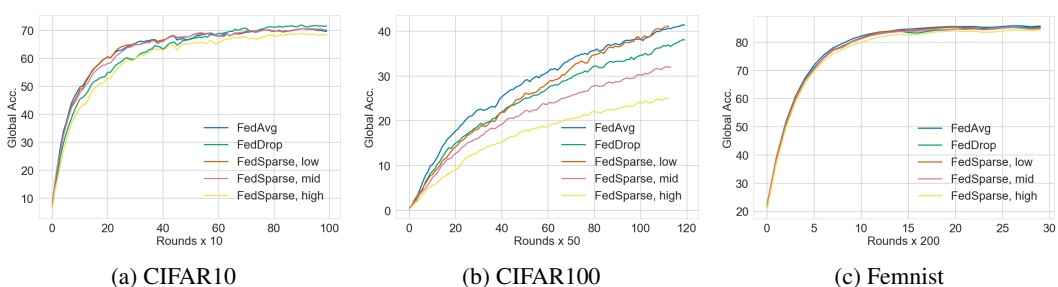

(a) CIFAR10                    (b) CIFAR100                    (c) Femnist

Figure 5: Evolution of the validation accuracy in terms of communication rounds.

**Impact of server side pruning.**    In order to understand whether server side pruning is harmful for convergence, we plot both the global and average local validation accuracy on CIFAR 10 for the "mid" setting of `FedSparse` with and without server side pruning enabled. As we can see, there are no noticeable differences and in fact, pruning results into a slightly better overall performance.

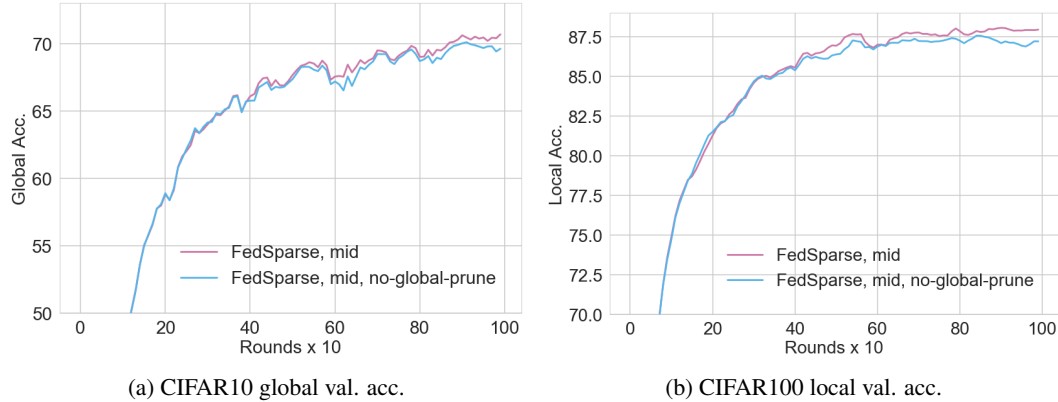

(a) CIFAR10 global val. acc.       (b) CIFAR100 local val. acc.

Figure 6: Evolution of the validation accuracy in terms of communication rounds with and without server side pruning.

## D  CORRESPONDENCE BETWEEN SINGLE STEP EM AND GRADIENT ASCENT

With the addition of the auxiliary variables $\phi_s$ we have that the overall objective for the server becomes

$$\arg\max_{\mathbf{w}} \frac{1}{S} \sum_{s=1}^{S} \log \int p(\mathcal{D}_s|\phi_s)p(\phi_s|\mathbf{w})\mathrm{d}\phi_s. \tag{21}$$

By performing EM with a single gradient step for $\mathbf{w}$ in the M-step (instead of full maximization), we are essentially doing gradient ascent on the original objective at 21. To see this, we can take the gradient of Eq. 21 w.r.t. $\mathbf{w}$ where $Z_s = \int p(\mathcal{D}_s|\phi_s)p(\phi_s|\mathbf{w})\mathrm{d}\phi_s$

$$\frac{1}{S} \sum_s \frac{1}{Z_s} \int p(\mathcal{D}_s|\phi_s)\frac{\partial p(\phi_s|\mathbf{w})}{\partial \mathbf{w}}\mathrm{d}\phi_s = \tag{22}$$

$$\frac{1}{S} \sum_s \int \frac{p(\mathcal{D}_s|\phi_s)p(\phi_s|\mathbf{w})}{Z_s}\frac{\partial \log p(\phi_s|\mathbf{w})}{\partial \mathbf{w}}\mathrm{d}\phi_s = \tag{23}$$

$$\frac{1}{S} \sum_s \int p(\phi_s|\mathcal{D}_s,\mathbf{w})\frac{\partial \log p(\phi_s|\mathbf{w})}{\partial \mathbf{w}}\mathrm{d}\phi_s \tag{24}$$

where to compute Eq. 24 we see that we first have to obtain the posterior distribution of the local variables $\phi_s$ and then estimate the gradient for $\mathbf{w}$ by marginalizing over this posterior.

## E  DERIVATION OF THE LOCAL LOSS

Let $p(\phi_{si}|\mathbf{w}_i,\mathbf{z}_{si}=1) = \mathcal{N}(\mathbf{w}_i,1/\lambda), p(\phi_{si}|\mathbf{w}_i,\mathbf{z}_{si}=0) = \mathcal{N}(0,1/\lambda_2)$ and $q(\phi_{si}|\mathbf{z}_{si}=1) = \mathcal{N}(\mathbf{w}_{si},\epsilon^2), q(\phi_{si}|\mathbf{z}_{si}=0) = \mathcal{N}(0,\epsilon^2)$. Furthermore, let $q(\mathbf{z}_{si}) = \mathrm{Bern}(\boldsymbol{\pi}_{si})$. The local objective that stems from 13 can be rewritten as:

$$\arg\max_{\mathbf{w}_{1:S},\boldsymbol{\pi}_{1:S}} \mathbb{E}_{q_{\boldsymbol{\pi}_s}(\mathbf{z}_s)q_{\mathbf{w}_s}(\phi_s|\mathbf{z}_s)}\big[\log p(\mathcal{D}_s|\phi_s)\big] - \mathbb{E}_{q_{\boldsymbol{\pi}_s}(\mathbf{z}_s)}\big[KL(q_{\mathbf{w}_s}(\phi_s|\mathbf{z}_s)||p(\phi_s|\mathbf{w},\mathbf{z}_s))\big]$$

$$+ \mathbb{E}_{q_{\boldsymbol{\pi}_s}(\mathbf{z}_s)}[\log p(\mathbf{z}_s|\boldsymbol{\theta})], \tag{25}$$

where we omitted from the objective the entropy of the distribution over the local gates.

One of the quantities that we are after is

$$\mathbb{E}_{q(\mathbf{z}_{si})}[KL(q(\phi_{si}|\mathbf{z}_{si})||p(\phi_{si}|\mathbf{z}_{si}))] =$$

$$\boldsymbol{\pi}_{si}KL(\mathcal{N}(\mathbf{w}_{si},\epsilon^2)||\mathcal{N}(\mathbf{w}_i,1/\lambda)) + (1-\boldsymbol{\pi}_{si})KL(\mathcal{N}(0,\epsilon^2)||\mathcal{N}(0,1/\lambda_2)). \tag{26}$$

The KL term for when $\mathbf{z}_{si}=1$ can be written as

$$KL(\mathcal{N}(\mathbf{w}_{si},\epsilon^2)||\mathcal{N}(\mathbf{w}_i,1/\lambda)) = -\frac{1}{2}\log\lambda - \log\epsilon + \frac{\lambda\epsilon^2}{2} - \frac{1}{2} + \frac{\lambda}{2}(\mathbf{w}_{si}-\mathbf{w}_i)^2. \tag{27}$$

The KL term for when $\mathbf{z}_{si} = 0$ can be written as

$$KL(\mathcal{N}(0, \epsilon^2)||\mathcal{N}(0, 1/\lambda_2)) = -\frac{1}{2}\log\lambda_2 - \log\epsilon + \frac{\lambda_2\epsilon^2}{2} - \frac{1}{2}. \tag{28}$$

Taking everything together we thus have

$$\mathbb{E}_{q(\mathbf{z}_{si})}[KL(q(\boldsymbol{\phi}_{si}|\mathbf{z}_{si})||p(\boldsymbol{\phi}_{si}|\mathbf{z}_{si}))] = \frac{\lambda\boldsymbol{\pi}_{si}}{2}(\mathbf{w}_{si} - \mathbf{w}_i)^2 + \boldsymbol{\pi}_{si}(-\frac{1}{2}\log\lambda - \log\epsilon + \frac{\lambda\epsilon^2}{2} - \frac{1}{2})+$$

$$(1 - \boldsymbol{\pi}_{si})(-\frac{1}{2}\log\lambda_2 - \log\epsilon + \frac{\lambda_2\epsilon^2}{2} - \frac{1}{2}) \tag{29}$$

$$= \frac{\lambda\boldsymbol{\pi}_{si}}{2}(\mathbf{w}_{si} - \mathbf{w}_i)^2 + \boldsymbol{\pi}_{si}\left(\frac{1}{2}\log\frac{\lambda_2}{\lambda} + \frac{\epsilon^2}{2}(\lambda - \lambda_2)\right) + C \tag{30}$$

$$\approx \frac{\lambda\boldsymbol{\pi}_{si}}{2}(\mathbf{w}_{si} - \mathbf{w}_i)^2 + \lambda_0\boldsymbol{\pi}_{si} + C, \tag{31}$$

where $\lambda_0 = \frac{1}{2}\log\frac{\lambda_2}{\lambda}$ and $\frac{\epsilon^2}{2}(\lambda - \lambda_2)$ was omitted due to $\epsilon^2 \approx 0$. In the appendix of Louizos et al. (2017), the authors argue about a hypothetical prior that results into needing $\lambda$ nats to transform that prior to the approximate posterior. Here we make this claim more precise and show that this prior is approximately equivalent to a mixture of Gaussians prior where the precision of the non-zero prior component $\lambda \to \epsilon$ (in order to avoid the $L_2$ regularization term) and the precision of the zeroth component $\lambda_2$ is equivalent to $\lambda \exp(2\lambda_0)$, where $\lambda_0$ is the desired $L_0$ regularization strength.

Furthermore, the cross-entropy from $q_{\boldsymbol{\pi}_s}(\mathbf{z}_s)$ to $p(\mathbf{z}_s|\boldsymbol{\theta})$ is straightforward to compute as

$$\mathbb{E}_{q_{\boldsymbol{\pi}_s}(\mathbf{z}_s)}[\log p(\mathbf{z}_s|\boldsymbol{\theta})] = \sum_j (\boldsymbol{\pi}_{sj}\log\boldsymbol{\theta}_j + (1 - \boldsymbol{\pi}_{sj})\log(1 - \boldsymbol{\theta}_j)). \tag{32}$$

By putting everything together we have that the local objective becomes

$$\underset{\mathbf{w}_s, \boldsymbol{\pi}_s}{\arg\max}\, \mathbb{E}_{q_{\boldsymbol{\pi}_s}(\mathbf{z}_s)}\left[\sum_i^{N_s} L(\mathcal{D}_{si}, \mathbf{w}_s \odot \mathbf{z}_s)\right] - \frac{\lambda}{2}\sum_j \boldsymbol{\pi}_{sj}(\mathbf{w}_{sj} - \mathbf{w}_j)^2 - \lambda_0\sum_j \boldsymbol{\pi}_{sj}$$

$$+ \sum_j (\boldsymbol{\pi}_{sj}\log\boldsymbol{\theta}_j + (1 - \boldsymbol{\pi}_{sj})\log(1 - \boldsymbol{\theta}_j)) + C. \tag{33}$$

## F   LOCAL OPTIMIZATION OF THE BINARY GATES

We propose to rewrite the local loss in Eq. 13 to

$$\mathcal{L}_s(\mathcal{D}_s, \mathbf{w}, \boldsymbol{\theta}, \boldsymbol{\phi}_s, \boldsymbol{\pi}_s) := \mathbb{E}_{q_{\boldsymbol{\pi}_s}(\mathbf{z}_s)}\left[\sum_i^{N_s} L(\mathcal{D}_{si}, \mathbf{w}_s \odot \mathbf{z}_s) - \lambda\sum_j \mathbb{I}[\mathbf{z}_{sj} \neq 0](\mathbf{w}_{sj} - \mathbf{w})^2\right.$$

$$\left. - \lambda_0\sum_j \mathbb{I}[\mathbf{z}_{sj} \neq 0] + \sum_j \left(\mathbb{I}[\mathbf{z}_{sj} \neq 0]\log\frac{\boldsymbol{\theta}_j}{1 - \boldsymbol{\theta}_j} + \log(1 - \boldsymbol{\theta}_j)\right)\right], \tag{34}$$

and then replace the Bernoulli distribution $q_{\boldsymbol{\pi}_s}(\mathbf{z}_s)$ with a continuous relaxation, the hard-concrete distribution (Louizos et al., 2017). Let the continuous relaxation be $r_{\mathbf{u}_s}(\mathbf{z}_s)$, where $\mathbf{u}_s$ are the parameters of the surrogate distribution. In this case the local objective becomes

$$\mathcal{L}_s(\mathcal{D}_s, \mathbf{w}, \boldsymbol{\theta}, \boldsymbol{\phi}_s, \mathbf{u}_s) := \mathbb{E}_{r_{\mathbf{u}_s}(\mathbf{z}_s)}\left[\sum_i^{N_s} L(\mathcal{D}_{si}, \mathbf{w}_s \odot \mathbf{z}_s)\right] - \lambda\sum_j R_{\mathbf{u}_{sj}}(\mathbf{z}_{sj} > 0)(\mathbf{w}_{sj} - \mathbf{w})^2$$

$$- \lambda_0\sum_j R_{\mathbf{u}_{sj}}(\mathbf{z}_{sj} > 0) + \sum_j \left(R_{\mathbf{u}_{sj}}(\mathbf{z}_{sj} > 0)\log\frac{\boldsymbol{\theta}_j}{1 - \boldsymbol{\theta}_j} + \log(1 - \boldsymbol{\theta}_j)\right), \tag{35}$$

where $R_{\mathbf{u}_s}(\cdot)$ is the cumulative distribution function (CDF) of the continuous relaxation $r_{\mathbf{u}_s}(\cdot)$. We can now straightforwardly optimize the surrogate objective with gradient ascent.

