# OpenReview forum: "Federated Averaging as Expectation Maximization"
_ICLR.cc/2021/Conference — Reject_

### Official Review · AnonReviewer1 · 2020-10-26
**two interesting observations; neither fully executed**

**Rating:** 5
**Confidence:** 4

**Review:**

Federated learning has emerged as a promising approach to training models at the edge devices. This paper makes an observation that most algorithms used within federated learning, including the popular FedAvg, could be cast as instances of EM methods. The paper then continues to propose FedSparse, a federated learning framework imposing sparse priors (specifically Bernoulli-Gaussian priors), and concludes with some experimental results on FedSparse.

The paper revolves around two disjoint stories: *1) Federated learning can be cast in the EM framework*; and *2) FedSparse can solve federated learning with sparse priors*. Unfortunately, neither of these two stories are told in a convincing manner. My personal suggestion to the authors would have been to focus on one of these messages and do a thorough work around it. Here are my questions on the different parts of the story for the authors to strengthen their story and contributions.

**1-a)** The equivalences between coordinate descent and full EM, and many of the other observations made in this paper only hold when we are dealing with strongly concave convex problems. The paper lacks a rigorous statement on what conditions are needed for these methods to converge and what convergence rate guarantees could be derived.

**1-b)** The observed equivalences in this paper are under the iid assumption. What happens in the presence of heterogeneous data?

**1-c)** Can you add a clear discussion on the connections between (Li et al. 2018), (Reddi et al. 2020), and the proposed EM framework in terms of the pros and cons and their suitability to different scenarios inspired by the observations in this paper? And can you then propose a set of experiments to verify such observations empirically?

**2-a)** How is FedSparse related to approximate message passing methods? AMP could also be cast as an instance of EM methods and is prevalent in solving sparse linear problems under the Bernoulli Gaussian prior assumption.

**2-b)** Can you please explain why the global accuracy of FedSparse is capped when compared to FedDrop? It is not clear to me why a sparse prior should be a good assumption in federated scenarios.

**2-c)** Can you please do a more in-depth study of the dependence of the performance of FedSparse in different scenarios? Can you also include (in the supplementary) a simulated scenario with truly sparse models and verify that FedSparse is well suited to such scenarios?

==== post rebuttal ====

It is an interesting observation that federated averaging and several other variants of it may be cast as instances of expectation maximization (EM). However, unfortunately, these connections have been made in a rather informal way, not resulting in much of new insights into convergence of such federated algorithms, or other practical tweaks to them. The paper then moves on to propose FedSparse, an abrupt shift from the original observation to federated learning with sparse priors. The study of FedSparse is thin, with unclear motivations, and with not so strong experimental results. Connections with the plethora of existing literature on sparse signal processing, specifically as it is cast in the EM framework, e.g., approximate message passing, is missing. In the end, the reader is left with two interesting ideas but it is not clear what the take-home message is.

My suggestion to the authors would be to "formalize" the connections between federated averaging and other instances of these algorithms (e.g., draw from EM literature to provide convergence guarantees, or maybe provide other variants of these algorithms with more favorable properties) and resubmit the paper with more formal connections to the EM framework. In my opinion, FedSparse is itself a separate paper that needs a separate motivation, set of hypotheses, and experiments.

---

> ### Author Response · Authors · 2020-11-16
> **Response to AnonReviewer1**
>
> We would like to thank you for the valuable feedback and pointers to improve our work. We believe that the two stories we discuss are complementary, in that we show how we can tackle practical problems in FL, namely reducing communication cost and learning efficient models, in a straightforward manner via the novel view of FedAvg that we provide. More specifically, it is due to the EM interpretation of FedAvg that we could easily develop the objective and updates of the FedSparse algorithm. We will update the text in order to make this more prominent. Now let us address your more specific questions:
>
> 1-a) For simplicity we will discuss this on a simpler model, namely $p(x| \theta) = \int p(z|\theta)p(x|z)dz$. Expectation-Maximization on this model can be formulated as coordinate ascent on an ELBO, i.e., a lower bound (akin to the one at eq. 9 in our submission) of the marginal likelihood $p(x|\theta)$, i.e., $L(\phi, \theta) = E_{q(z| \phi)}[log p(x|z) + log p(z|\theta) – log q(z| \phi)]$, where we alternate between optimizing $\phi$ and $\theta$. The conditions for this to hold is that a) the approximating distribution has enough flexibility such that it can recover the true posterior and b) there is a closed form solution for the M-step for the model parameters $\theta$. In this case, maximizing the ELBO with respect to $\phi$ will result into $q(z|\phi) = p(z|x, \theta) = p(z|\theta)p(x|z) / p(x|\theta)$, i.e. it will recover the true posterior and will correspond to the E-step. This is irrespective of the nature of the likelihood. Then by keeping $\phi$ fixed and maximizing with respect to $\theta$ we see that the only term that involves $\theta$ is $F=E_{q(z|\phi)}[log p(z|\theta)]$. We can then either do “full EM” if we can solve $dF/d\theta = 0$ for $\theta$ or “Generalized EM” if we make a single gradient step for $\theta$. In the latter case, this is equivalent to a single gradient step on the marginal likelihood as we show in Appendix C. It should be noted that both approaches are valid, since they both increase the marginal likelihood at each step and both can converge to a (local) maximum. Having said that, perfect optimization for $\phi$ (i.e., inference for the E-step) might not be possible, thus there will be some bias in the optimization procedure. Nevertheless, in practice this does not seem to be a big problem given the successes of variational inference in deep learning.
>
> 1-b) We do not make any i.i.d. assumption in the paper and simply follow the assumptions that FedAvg does, i.e., that the shards/clients are independent conditioned on the server parameters $w$. In fact, all of our experiments are in the heterogeneous setting, since the datasets are split in a non-i.i.d. fashion.
>
> 1-c) Both Li et al. and Reddi et al. are specific instances of the same EM framework we propose. Li et al. uses a non-trivial $\lambda$ for the prior $p(\phi_s | w)$ at the shard level and performs a full M-step, i.e., parameter averaging. In contrast, Reddi et al. corresponds to $\lambda \rightarrow 0$ and performing a single gradient step for the M-step. In practice, we find that the Reddi et al. approach performs better than Li et al. in that (a) the drift of $\phi_s$ from $w$ is implicitly controlled by the small number of local updates before communicating and (b) doing a gradient step for the M-step allows us to use adaptive optimizers such as Adam.
>
> 2-a) FedSparse is an instance of a method that performs MAP inference for the E-step. Instead of this any other message passing method can be used in order to obtain either a point estimate or an approximation to the posterior. We chose MAP inference due to it being simpler to work with in complex non-linear models such as neural networks.
>
> 2-b) Could you perhaps elaborate what you mean here? We do not cap any accuracy metric. A sparse prior is beneficial from a practical perspective in FL; we can use it to reduce communication costs and we can also use it to learn more efficient models (as they will be executed on resource constrained edge devices, such as mobile phones). We would like to refer you to a recent review paper for FL https://arxiv.org/abs/1912.04977, section 3.5.
>
> 2-c) Could you elaborate a bit more, what you mean here? We experimented with different regularization strengths on all three of the datasets in order to show how FedSparse behaves on each scenario. The objective of FedSparse is to find a reasonably good model (accuracy wise) that is efficient and not necessarily to recover the true underlying model (which may or may not be sparse in the case of neural networks). The former is possible in general with neural networks, as it has been shown that they can be pruned quite significantly without large degradations in performance.
>
> We hope that we have answered your questions sufficiently and that you will consider re-evaluating our contribution. Let us know if there are still things that are unclear.

---

### Official Review · AnonReviewer3 · 2020-10-27
**The paper has a novel idea but needs more clarification.**

**Rating:** 5
**Confidence:** 4

**Review:**


The paper tries to use EM to explain the optimization procedure of the training method in a federated setting, and then treats the local model as a hidden variable of EM. Then, the authors propose a new FedSparse method to reduce communication efficiency by training a sparse model.

Pros:

It is a novel idea to treat local mode’s parameter \phi as a hidden variable in an EM framework. In particular, the following FedSparse algorithm is designed based on this setting.

This proposed sparsity-based model compression method is linked to federated learning in a comprehensive way.

As compared to FedAvg, the proposed method can reduce about 50% communication cost without scarifying accuracy.

Cons:

The FedAvg is an SGD-based optimization, and treating the local model’s parameters as hidden variable phi_s is a little bit confusing. In particular, the FedAvg has no regularization term to prevent \phi moving too far from w, and the FedAvg’s variant version involves such a mode divergence-based regularization term – FedProx (Tian Li et. al. 2018).  Therefore, the definition of p(phi | w) in Equation 3 need more discussion.

The paper uses a classic sparsity method, spike, and slab (Mitchell & Beauchamp, 1988). The authors need to introduce a more model sparsity method for compression purposes and then justify choosing the spike and slab.

In the experiment part, the proposed method and baseline methods have no significant difference in terms of accuracy. From a communication efficiency perspective, the proposed method should be also compared to FedDrop. Moreover, reduced communication cost in FedSparse is not very high.

---

> ### Author Response · Authors · 2020-11-16
> **Response to AnonReviewer3**
>
> Thank you for the encouraging comments regarding the novelty of our framework and the effectiveness of FedSparse. We will now address your comments.
>
> Regarding the $p(\phi_s | w)$ term; indeed while there is no explicit such term in FedAvg, we still argue that the incomplete local optimization from FedAvg implies a specific prior that prevents $\phi_s$ from moving too far from $w$. In fact, for linear regression this can precisely manifest as a Gaussian prior centered around the initial point, see Santos 1996 reference in our submission. For non-linear models it is more complex, but there is still an implicit bound on the distance. Having said that, this reasoning becomes looser when we optimize the local model for longer at the client level (e.g., more than E=1) and in such cases, an additional term akin to the one from FedProx might be necessary for better averaging.  Empirically, on our experiments it didn’t matter whether we used an additional FedProx term for E=1.
>
> Regarding the sparsity method; could you perhaps elaborate on what you mean with a “more model sparsity method” for compression? We chose the spike and slab, due to the nice properties of decomposing sparsity (modelled with the spike) from parameter shrinkage (modelled with the slab). Having said that, alternative approaches with perhaps different priors are interesting and could be explored in future work.
>
> Regarding the experiments; we do compare our proposed method, FedSparse, with FedDrop on all of the tasks we considered and found overall better performance. Furthermore, the reduction in communication cost would be relative to the amount of performance degradation we are willing to allow. Having said that, FedSparse is orthogonal to methods that employ, e.g., quantization and gradient sparsification, and could be combined with them in order to achieve even more savings. We left such an exploration for future work.
>
> We hope that our response addresses your concerns sufficiently. Let us know if there is anything else that is unclear.

---

### Official Review · AnonReviewer2 · 2020-10-30
**An interesting framework, but needs more clarifications and analysis, either theoretical or in the form of a careful ablation study**

**Rating:** 4
**Confidence:** 4

**Review:**

**Summary**

The paper proposes to re-interpret federated averaging (FedAvg) as a version of the expectation-maximization (EM) algorithm under a particular probabilistic model. Further, the authors propose to use spike-and-slab sparsity inducing priors over the local model parameters to sparsify the learned models (the corresponding method is called FedSparse), which naturally reduces the communication cost (only non-zero parameters need to be sent over the network). Improvement in communication efficiency is showcased on a few standard federated datasets.

**General comments and evaluation**

In my opinion, the paper has some interesting ideas; the effort to connect federated learning with techniques from probabilistic inference strongly resonates with me. However, the connection the authors make between federated averaging and EM is not as insightful as I anticipated. The fact that FedAvg can be seen as is a special case of EM with a particular choice of a prior distribution does not add much to our understanding of the behavior of the algorithm (e.g., convergence, issues related to different types of heterogeneity, etc.). While FedSparse is derived using Bayesian considerations, I would argue that the same algorithm can be derived using FedAvg + simple L1 regularization. Again, I do not see the benefit of the Bayesian formalism, especially given that it is essentially thrown away at the end by using the "hard" version of EM (where all the Gaussian distributions are collapsed). This unnecessarily complicates things without offering much insight.

Apart from the formalism, the authors should clarify the modeling assumptions of the FL setting they consider. Currently, there are two canonical FL settings, known as "cross-device" and "cross-silo" (https://arxiv.org/abs/1912.04977), which require FL methods to operate under drastically different assumptions -- the former assumes that the data is distributed among a very large number of devices (tens of millions) with limited memory and compute; the latter assumes a small number of clients with sufficient computational resources. It is unclear which of the settings the authors aim to address. For instance, $\arg\max_{w_{1:S}}$ in Eq. 13 seems meaningless in the cross-device setting since in practice S would be on the order of millions and only a subset of clients will participate in each round. I'm curious to hear which setting the authors had in mind.

All in all, I think the paper would significantly benefit from further clarification, a possibly theoretical analysis of the algorithm in heterogeneous settings (e.g., understand convergence rate and bounds on the tradeoffs between sparsity and performance) and/or a more careful and convincing empirical study.

**Detailed comments and questions**

- While reading the paper, a high-level question that wasn't clear to me: why use specifically "hard" EM for FedSparse instead of the standard EM?

- In the paragraph after Eq. 13, the authors propose a bunch of approximations, starting with collapsing Gaussian distributions, then removing the entropy term from the objective, setting hyperparameters, etc. This is a lot of approximations, I would like to know the purpose and the effect of each on the final result. Currently, many of these choices are obscure and unclear.

- The authors propose to use sampling of the binary variables z to compute gradients in Eqs. 18-19. The approach is indeed unbiased, but I'm worried that the increase in variance might be substantial in some practical cases. Can the authors somehow quantify this? (Ideally, provide a convergence rate for the algorithm that shows the effect of additional variance).

- What is the effect of structured pruning proposed to reduce server-to-client communication cost? It seems like a heuristic and it is unclear how it affects the algorithm.

**Experiments**

- Could the authors define local vs. global accuracy exactly?
- How many clients participated in each round in each benchmark? Was the server communicating with all clients or just a subset?
- The authors show some improvement of FedSparse over FedAvg and FedDrop in terms of the total amount of communication traffic in GB. Why does that metric matter? In which use-case? This goes back to my question about modeling assumptions.
- How many epochs per round did each client run? I would be curious to see how FedSparse compares to FedAvg if we double the amount of local computation (eg, make clients train for 2 epochs at each round instead of 1). I imagine that would improve convergence and reduce # of rounds, and as a result, reduce the total communication cost.
- I think that the benchmark datasets considered in the study may not be the best to showcase the benefit of sparsity. I would be curious to see how FedSparse compares to FedAvg on StackOverflow dataset introduced by Reddi et al. (2020), where the sparsity benefits can be much more visible. Eg, learning extremely sparse models could enable the use of much larger vocabularies, and as a result, produce more accurate models than what is achievable with FedAvg. That would be much more compelling evidence in favor of FedSparse than the results reported in the paper.

---

> ### Author Response · Authors · 2020-11-16
> **Response to AnonReviewer2 [1 / 2]**
>
> We would like to thank you for the thorough and well-thought review! Let us address your concerns.
>
> First of all, we believe that unifying views are interesting in that they can generate new insights and research. This is something that has happened in the past, e.g., with the seminal work https://www.mitpressjournals.org/doi/abs/10.1162/089976699300016674.
> In this sense, the EM view of FedAvg helps in making the modelling assumptions of FedAvg clearer, i.e., a hierarchical model with Gaussian priors. The main benefit in this case is extendibility. We can easily extend FedAvg in novel ways by appropriately choosing the prior that the server has; FedSparse was one case where we used a Spike and Slab prior to induce sparsity but is certainly not the only one. Alternative priors would be a Laplace prior which would select the median parameter instead of averaging or a mixture of Gaussians prior which would create a collection of models at the server by performing clustering at the client weights. FedAvg + L1 regularization could similarly be derived from appropriate choice of priors in both the server and the client. We will add a couple of sentences about this in the main text.  Overall, we opted for hard-EM because (a) we wanted to stay close to the FedAvg paradigm of simple point estimation and (2) approximate inference, such as variational inference or stochastic gradient MCMC, is more involved in order to get it right for complex posteriors, such as those that arise in neural networks. Having said that, we did not explore it too deeply and it is definitely something that we consider for future work. There are a lot of interesting ways in how full inference can benefit FL, e.g., for representing uncertainty in the predictions which is especially important for safety-critical FL or cases where clients have a very small amount of data.
>
> As for the modelling assumptions of the FL setting; we are operating on the cross-device setting and we will update the draft to reflect that. Notice that in general we can always approximate the loss and hence server-side gradients / stationary points (i.e., averages) by using only a subset of the devices at each round, as is the case for vanilla FedAvg. In fact this is the setting we used in all of our experiments, where we randomly sampled 10 clients without replacement for each round (a common setting in FL) in order to compute the gradient. We will make this clearer in the text and add a description of the subsampling. Furthermore, we would also like to state that all of our experiments are in the heterogeneous setting, since on all datasets we used non-iid splits for the data. For more details we refer to the second paragraph of section 5.
>
> Regarding the approximations from eq.13; these approximations stem from only one assumption; that of doing hard-EM. Since we are looking for point estimates, we do not need any entropy terms or variances for the local parameters. The final objective function that we end up with is simple and intuitive, since it performs L0 regularization at the client level and simple (weighted) averaging at the server (if we opt for stationary points instead of gradients).
>
> Regarding sampling the z variables; empirically we see that this does not affect the convergence speed of the algorithm, as usually FedSparse converges at the same rate as FedAvg (depending on the sparsity). We will add a plot of this in the Appendix. As for the structured pruning; it does not affect the performance, as those parameters are mostly zero anyway. We performed an experiment where we do not perform pruning for the server to client communication, and we did not see differences in performance. We will add this plot in the Appendix.

---

> > ### Author Response · Authors · 2020-11-16
> > **Response to AnonReviewer2 [2 / 2]**
> >
> > As for the experimental setup questions;
> > -	the global accuracy is the accuracy of the global model, i.e., the one that uses $\theta \odot w$ for the parameters of the network, on the union of the shard specific test sets. The local accuracy is the average accuracy of the locally fine-tuned models, i.e., the ones that use $\pi_s \odot w_s$, on their shard specific test sets (i.e., test sets that have the same distribution of labels as their respective training sets). These locally fine-tuned models are the ones that each client last communicated to the server.
> > -	Regarding the GB metric; we would like to point out that reducing the communication costs is an active research area in FL as it is important when considering real world applications of FL. Current practices employ FL only when a phone is charging and is on a WiFi network, but by reducing the communication costs through e.g. sparsifying the model we could move towards a setting where FL could also happen on battery and a mobile network. In fact, joint tackling of communication cost reduction and learning an efficient model was also identified as a promising research direction in https://arxiv.org/abs/1912.04977 (section 3.5). We will add this as a motivation in the paper.
> > -	We used an E=1 for all methods and datasets as this is usually common practice in FL works. We would like to point out that more local epochs do not necessarily improve convergence, especially in the case of non-iid data (due to the larger drift and potential disagreement of the local models).
> > -	As for alternative datasets; your stackoverflow suggestion is an interesting choice in case one seeks to employ sparsity for better performance of the final model. Our main motivation was sparsity for inference and communication efficiency, as a client will need both less computation in order to make predictions (and thus reduce battery consumption) and less communication due to sending only a subset of the model to the server. For the latter, we believe that our experiments highlight the benefits of FedSparse.
> >
> > We hope that we answered sufficiently your questions and that you would consider raising your score. If anything is still unclear, please let us know so that we clarify more.

---

### Official Review · AnonReviewer4 · 2020-10-30
**In overall, this paper adds valuable insights into designing fedearated optimization methods with help of EM analysis. I believe this paper adds decent contributions.**

**Rating:** 7
**Confidence:** 2

**Review:**

Summary of the paper:
This paper brings into light a new perspective where the well known "FedAvg" algorithm can be modeled as expectation maximization. This reveals certain characteristics about federated averaging which can be useful in practice. Using these insights, "Fedsparse" algorithm is introduces that provides a trade-off between the sparsity (thus the model size) and accuracy of the models. Ample experiments are provided to validate theoretical claims.

Quality:
Presentation and the quality of writing is decent and the flow is understandable. But the major contributions are not clearly stated. I am wondering why the related work is at the end. I think it is better move the related work to an early section along with the emphasis on contributions.

Originality and significance:
I believe the connection between FedAvg and EM and the related understandings open up new avenues for designing better federated learning algorithms. FedSparse is clearly one of such methods that allows a trade-off between accuracy and memory consumption. Overall, believe this paper adds valuable contributions to the broader ICLR community.

Other comments:
When communicating the parameters from server to clients, I am wondering whether there are certain heuristics that can be used to prune $\theta$ values. Using an arbitrary threshold may not be the best given that the heterogeneous and non I.I.D. local datasets. Perhaps the authors can look into the effects of this as a future direction.

==============================================================================================================

Added after reading author response:

-----------------------------------------------------------------------

Authors have sufficiently addressed my concerns and I'm planning to maintain my generous score based on my initial understanding. However, other reviewers have raised many important concerns and I encourage authors to improve the paper based on those.

---

> ### Author Response · Authors · 2020-11-16
> **Response to AnonReviewer4**
>
> Thank you for the encouraging words and review. We updated the paper in order to make the contributions more prominent and we also state them below for completeness.
> -	We derive the popular Federated Averaging algorithm though a hard version of Expectation-Maximization on a simple hierarchical model. This view of FedAvg has not been considered before and we believe it allows for novel and useful extensions of FedAvg; FedSparse being one of them. We also connect several recent works such as [Li et al. 2018, Reddi et al.] under one framework and show how FedAvg connects to meta-learning.
> -	We introduce FedSparse, an algorithm that can learn sparse neural networks in the federated setting. This has two key advantages; 1. It reduces the communication cost during training and 2. It reduces the computational complexity of inference at resource constrained edge devices, which can result into less power consumption and faster inferences.
>
> As for alternative ways to prune; indeed, this is something worthwhile to explore. For example, each client can prune the model according to the local distribution $q(z_s)$ and ask from the server only that subset. This is something that we will explore for future work.
>
> Regarding the exact heuristic (i.e., thresholding) we used for pruning in the server to client communication; in practice we find that the accuracy does not change if we do not do any thresholding for the server to client communication. As a result, this specific heuristic is not harmful performance wise and it further reduces the communication costs.
>
> We hope that our response can further reinforce your confidence in our contributions.

---

### Author Response · Authors · 2020-11-24
**The discussion phase is almost over**

As the discussion period is coming to an end, we are thankful for the reviewers’ time and hope that they had a chance to read our rebuttal. If you feel like we adequately addressed your concerns, we would be thankful for a comment and a reevaluation of the current form of our submission.

---

### Decision · Program_Chairs · 2021-01-07
**Final Decision**

**Decision:**

Reject

**Comment:**

The reviewers agree that the EM perspective of Federated Learning is novel and interesting. However, a common criticism is that the connection made is rather shallow and not sufficiently developed. There look to be quite interesting potentials of the proposed framework and the specific FedSparse method, but I agree with the reviewers that both aspects need further development before they are in publishable form.